# Human subthalamic nucleus activity during non-motor decision making

**Baltazar A Zavala, Anthony I Jang, Kareem A Zaghloul***

Surgical Neurology Branch, National Institute of Neurological Disorders and Stroke, Bethesda, United States

**Abstract** Recent studies have implicated the subthalamic nucleus (STN) in decisions that involve inhibiting movements. Many of the decisions that we make in our daily lives, however, do not involve any motor actions. We studied non-motor decision making by recording intraoperative STN and prefrontal cortex (PFC) electrophysiology as participants perform a novel task that required them to decide whether to encode items into working memory. During all encoding trials, beta band (15–30 Hz) activity decreased in the STN and PFC, and this decrease was progressively enhanced as more items were stored into working memory. Crucially, the STN and lateral PFC beta decrease was significantly attenuated during the trials in which participants were instructed not to encode the presented stimulus. These changes were associated with increase lateral PFC-STN coherence and altered STN neuronal spiking. Our results shed light on why states of altered basal ganglia activity disrupt both motor function and cognition.
DOI: https://doi.org/10.7554/eLife.31007.001

## Introduction

The basal ganglia have traditionally been implicated in facilitating or inhibiting motor movements (*Albin et al., 1989*; *DeLong, 1990*). As such, studies tying the basal ganglia to decision making have focused on decisions in which participants either make or withhold a motor command (*Zavala et al., 2015*). Yet, many of the choices that we make in our daily lives involve decisions that do not explicitly require an immediate motor action. For instance, when meeting someone at a dinner party, one may decide that their name is worth trying to remember. Whether and how the basal ganglia may also participate in such non-motor decisions remains poorly understood.

Given their widespread connections to cortical regions, we hypothesized that the circuits and structures that enable the basal ganglia to mediate motor decisions may also be used in an analogous manner to participate in non-motor decisions. We were specifically interested in the non-motor decision to attend to and encode items into working memory. Our ability to dynamically control access to working memory is critical as only a small portion of the large number of stimuli we are exposed to each day are relevant for our behavioral goals. Most studies of working memory have implicated several regions in the prefrontal cortex (PFC) in regulating whether and how memories are successfully encoded (*McNab and Klingberg, 2008*; *Spitzer et al., 2010*; *Voytek and Knight, 2010*; *Heinrichs-Graham and Wilson, 2015*; *Wiesman et al., 2016*). We hypothesized, however, that gating access to working memory, like gating movement, also involves the basal ganglia.

Several lines of evidence suggest this possibility. The decision to execute or withhold a movement involves a race between competing basal ganglia pathways that facilitate or inhibit motor commands (*Schmidt et al., 2013*). At the center of this race lies the subthalamic nucleus (STN), a structure within the basal ganglia previously implicated in motor decisions (*Kühn et al., 2004*; *Frank et al., 2007*; *Cavanagh et al., 2011*; *Zaghloul et al., 2012*; *Alegre et al., 2013*; *Zavala et al., 2014*). The ability to gate movement may also grant the basal ganglia, and the STN, the ability to regulate non-motor cognitive processes such as memory. In addition, the basal ganglia exhibit extensive

*For correspondence:
kareem.zaghloul@nih.gov

**Competing interests:** The authors declare that no competing interests exist.

**eLife digest** Should I run to catch the bus, or wait for the next one? Deciding whether or not to move may be a common part of our life, but many decisions we make every day do not involve any physical activity, such as deciding whether something is important or not. But does the brain distinguish between decisions involving movement and those without? Previous research has shown that a cluster of neurons called the basal ganglia and a region within these clusters called the subthalamic nucleus play a critical role in both movement and decision-making that involves activity.

The basal ganglia are widely connected to other regions in the brain such as the prefrontal cortex, which is important for decision making and forming associated memories. Therefore, scientists think that the basal ganglia may also play a role in making decisions that do not involve movement, such as deciding whether to pay attention and form relevant memories. Indeed, in patients with Parkinson's disease – a condition that damages parts of the brain and affects their movement – the degree of physical impairment correlates with their memory deficits.

To test this, Zavala et al. recorded the activity of the subthalamic nucleus and the prefrontal cortex of patients with Parkinson's disease while they performed a task that required them to decide whether to remember something or not. The recordings were taken as the patients underwent 'deep brain stimulation surgery', which is used to treat the symptoms of Parkinson's.

The results showed that brain activity in both regions decreased during the task. This is what normally happens when people make movements, but in this case, these decreases occurred even when individuals were not moving and instead just formed memories. In addition, individuals were presented with specific items to remember and others to forget. When they were presented with items to forget, the activity rebounded back to its original levels. Similar activity patterns have also been observed when individuals decided to stop a movement.

This confirms that the subthalamic nucleus plays a role in decision-making and shows that this area is involved in decisions, even when they do not involve a movement. A full understanding of the purpose of the subthalamic nucleus and basal ganglia will help us understand why changes in the activity of basal ganglia leads to the memory and movement deficits seen in Parkinson's disease.
DOI: https://doi.org/10.7554/eLife.31007.002

connections with regions of the PFC that have been implicated in working memory (*Alexander et al., 1986*; *Nakano et al., 2000*; *Chudasama et al., 2003*), and imaging studies reveal increased basal ganglia activity during tasks that involve working memory (*Chang et al., 2007*; *McNab and Klingberg, 2008*; *Murty et al., 2011*; *Chatham et al., 2014*). Empiric support for the basal ganglia's role in gating memory also arises from studies of patients with Parkinson's disease who have disruptions of normal basal ganglia circuitry (*Hammond et al., 2007*). These patients often experience working memory deficits that correlate with the degree of motor and executive function impairment (*Owen et al., 1997*; *Lewis et al., 2003*; *Moustafa et al., 2008*; *Chiaravalloti et al., 2014*; *Trujillo et al., 2015*; *Wiesman et al., 2016*). Moreover, patients receiving deep brain stimulation (DBS), in which high-frequency electrical stimulation is delivered directly to the STN, may experience changes in working memory performance, although studies examining whether DBS improves or worsens working memory have not been conclusive (*Hälbig et al., 2004*; *Hershey et al., 2008*; *Mayer et al., 2016*; *Ventre-Dominey et al., 2014*; *Mollion et al., 2011*).

Collectively, these studies suggest a link between the basal ganglia and non-motor decisions such as those used to regulate and access working memory. Despite such empiric support, however, few studies have directly examined neural activity of the human basal ganglia, and specifically the STN, during a working memory task. We address this here by examining local field potential and neuronal spiking activity in the human STN in patients undergoing DBS surgery for Parkinson's disease as they participate in a task requiring them to make non-motor decisions to selectively attend to and encode items into working memory. Because of the known involvement of the PFC in working memory, and because of the known connections between the PFC and the basal ganglia, we also simultaneously capture intracranial EEG (iEEG) activity from the PFC in order to ask how these structures communicate during this process of making non-motor decisions. As such, our work builds upon previous studies establishing the role of the human STN in motor response inhibition and

decision making (*Zavala et al., 2015*) to demonstrate the STN's involvement in non-motor decisions used to dynamically modulate working memory.

## Results

Eighteen participants (16 males; 58.4 ± 1.64 (mean ± SEM) years old) undergoing DBS surgery for Parkinson's disease performed a novel working memory task in which we sequentially presented eight single-digit numbers (*Figure 1A*; see Materials and methods). We instructed the participants to attend to and memorize only numbers presented within a target shape (square or octagon), which was pseudo-randomly chosen and presented before each block of eight numbers. During each block, four of the numbers appeared within the target shape (target trials). The target numbers were randomly interleaved with four numbers appearing within a distractor shape (distractor trials). Correct performance in each block required the participant to vocalize only the four target numbers at the end of each block. Participants vocalized the correct response on 71.2 ± 2.6% of the blocks they were shown. In the remaining 28.8 ± 2.6% blocks, the participants made one of two types of errors: they could fail to vocalize all four target numbers (96.4 ± 1.2% of error blocks) or they could erroneously include a distractor number in the vocalized sequence (63 ± 4.5% of error blocks). On 62.1 ± 4.4% of the error blocks, they made both types of errors.

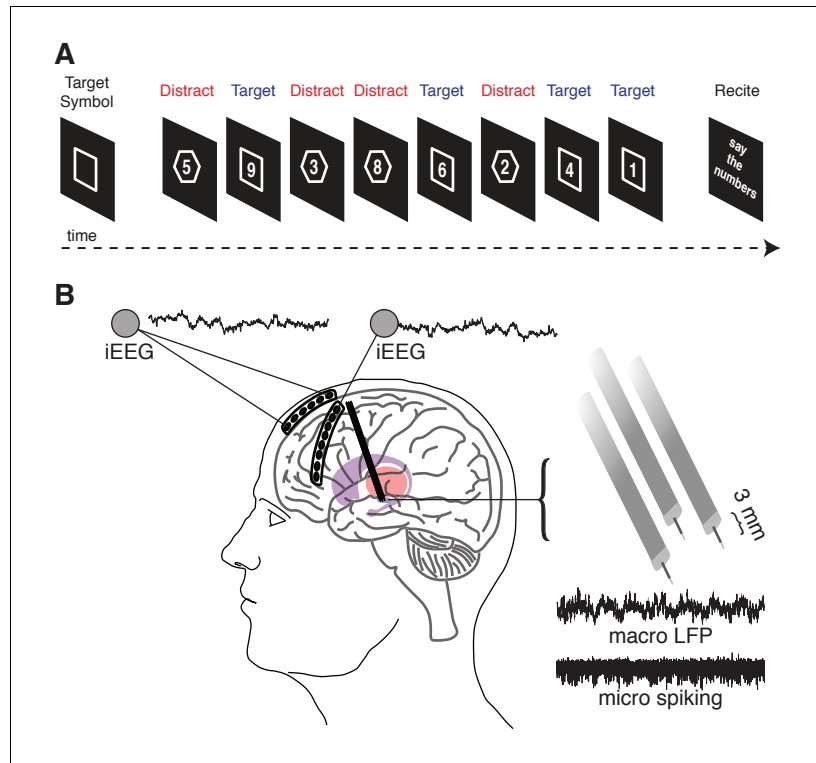

**Figure 1.** Memory task during recordings captured from the human STN and cortex. (**A**) A target shape (square or octagon) appears on the screen to indicate the target shape for the upcoming block. The participant then pushes a button to begin the block. Eight numbers then appear sequentially, each within either a square or an octagon. The participant is instructed to remember only the numbers appearing within the target shape (target trials) and ignore the remaining numbers (distractor trials). At the end of the block, the participant verbalizes the four target numbers. (**B**) Six- and eight-contact electrode strips are placed over the anterior and lateral PFC, respectively, to capture intracranial EEG activity during the task. Simultaneously, three macro/microelectrode pairs capture LFP (macro) and action potential spiking activity (micro) from the STN. A 5 s sample recording is shown for each.
DOI: https://doi.org/10.7554/eLife.31007.003

The following figure supplement is available for figure 1:

**Figure supplement 1.** iEEG electrode locations.
DOI: https://doi.org/10.7554/eLife.31007.004

We simultaneously captured micro- and macroelectrode recordings from the subthalamic nucleus (STN), and intracranial EEG recordings from subdural electrodes temporarily placed over the anterior and lateral prefrontal cortex (PFC) as participants performed the task (*Figure 1B* and *Figure 1—figure supplement 1*). We first investigated overall changes in power during all trials in each of three recorded brain regions - the anterior PFC, the lateral PFC, and the STN - by computing the average power across all trials during each experimental session and across all electrodes in each brain region (*Figure 2A,B*). Across sessions, all three brain regions demonstrated significant decreases in beta band (15–30 Hz) power across all trials (p < 0.005, permutation test; see Materials and methods). We also found significant increases in theta band (2–8 Hz) power in the anterior and lateral PFC during the task (p < 0.005 for anterior PFC and lateral PFC, permutation test, *Figure 3—figure supplement 1A*). These changes were not specific to either left- or right-sided brain regions, as we found no significant differences in spectral power between experimental sessions captured in the separate hemispheres (left anterior PFC vs right anterior PFC $p = 0.37$, left lateral PFC vs right lateral PFC $p = 0.55$, left STN vs right STN $p = 0.60$, permutation test; see Materials and methods).

Although we observed these decreases in beta oscillatory power in both the anterior and lateral PFCs by averaging the spectral power across all electrode contacts, we were interested in whether the observed changes represented a focal process or a diffuse cortical phenomenon. We therefore examined changes in spectral power within each electrode contact. In 19 out of the 24 sessions with lateral PFC electrodes, at least one electrode contact exhibited a significant decrease in beta power during the task (p < 0.05, permutation test; see Materials and methods). This effect was not present in all electrode contacts, however. Indeed, within these 19 sessions, we found that only $2.6 \pm 0.29$ of seven lateral PFC electrode contacts ($36.8 \pm 4.1\%$) showed a significant decrease in beta power relative to baseline (p < 0.05, permutation test; *Figure 2—figure supplement 1*). In contrast, 14 out of 29 sessions with anterior PFC electrodes demonstrated at least one electrode contact exhibiting a significant decrease in beta power during the task (p < 0.05, permutation test). Within these sessions, we found that $2.1 \pm 0.31$ of five electrode contacts ($42.9 \pm 6.2\%$) showed the change in beta power relative to baseline (p < 0.05, permutation test). Although we did not have access to intraoperative computed tomography (CT), we estimated all anterior and lateral PFC electrode locations on each participant's brain (see Supplementary Information) to determine whether the observed effects were localized to a particular brain region. We found that despite the fact that the anterior PFC, the lateral PFC, and the STN all showed a similar drop in beta power during the task, the beta power within individual participants was focal in nature, although the location of these focal drops was not consistent across participants (*Figure 2—figure supplement 1*).

We were interested in understanding how the observed changes in beta oscillatory power evolved over the course of successive trials in each individual block. Although each individual trial exhibited a relative transient decrease in beta power, overall beta power exhibited a systematic decrease from the first through the last trial of each block in all three brain regions (*Figure 2C*). The observed decreases across trials were not related to movement as participants were instructed to withhold all movements until receiving an instruction to vocalize their response one second after the final number was cleared from the screen. We quantified this overall decrease by examining beta oscillatory power averaged across the entire trial (0 to 1000 ms following stimulus onset). A repeated-measures ANOVA revealed a significant effect of trial number on the changes in average beta oscillatory power in all three brain regions (anterior PFC $F = 4.88$, $df = 7$, p < 0.001; lateral PFC $F = 6.46$, $df = 7$, p < 0.001; STN $F = 4.07$, $df = 7$, p < 0.001). We also performed a linear regression of beta power against trial number and found a significant relation across all experimental sessions in all three brain regions (anterior PFC $\rho = -0.31 \pm 0.1$, $t(27) = 3.4$, p < 0.002; lateral $\rho = -0.31 \pm 0.1$, $t(23) = 3.1$, p < 0.005; STN $\rho = -0.35 \pm 0.1$, $t(28) = 3.35$, p < 0.002; best fit line in *Figure 2C*). See Supplementary Data and *Figure 3—figure supplement 3* for analysis of progressive decrease in beta oscillatory power separately for target and distractor trials.

Our data suggest that participation in this non-motor task evokes decreases in beta oscillatory power in both the STN and PFC following the presentation of each number. The observed decreases in beta oscillatory power during each trial are similar to decreases previously observed with motor movements (*Kühn et al., 2004*; *Ray et al., 2012*; *Alegre et al., 2013*). However, we were specifically interested in whether these changes in beta oscillatory power depend on whether the presented stimulus was a target or distractor item. We therefore investigated whether there were significant differences in oscillatory power between the target and distractor conditions. We computed the

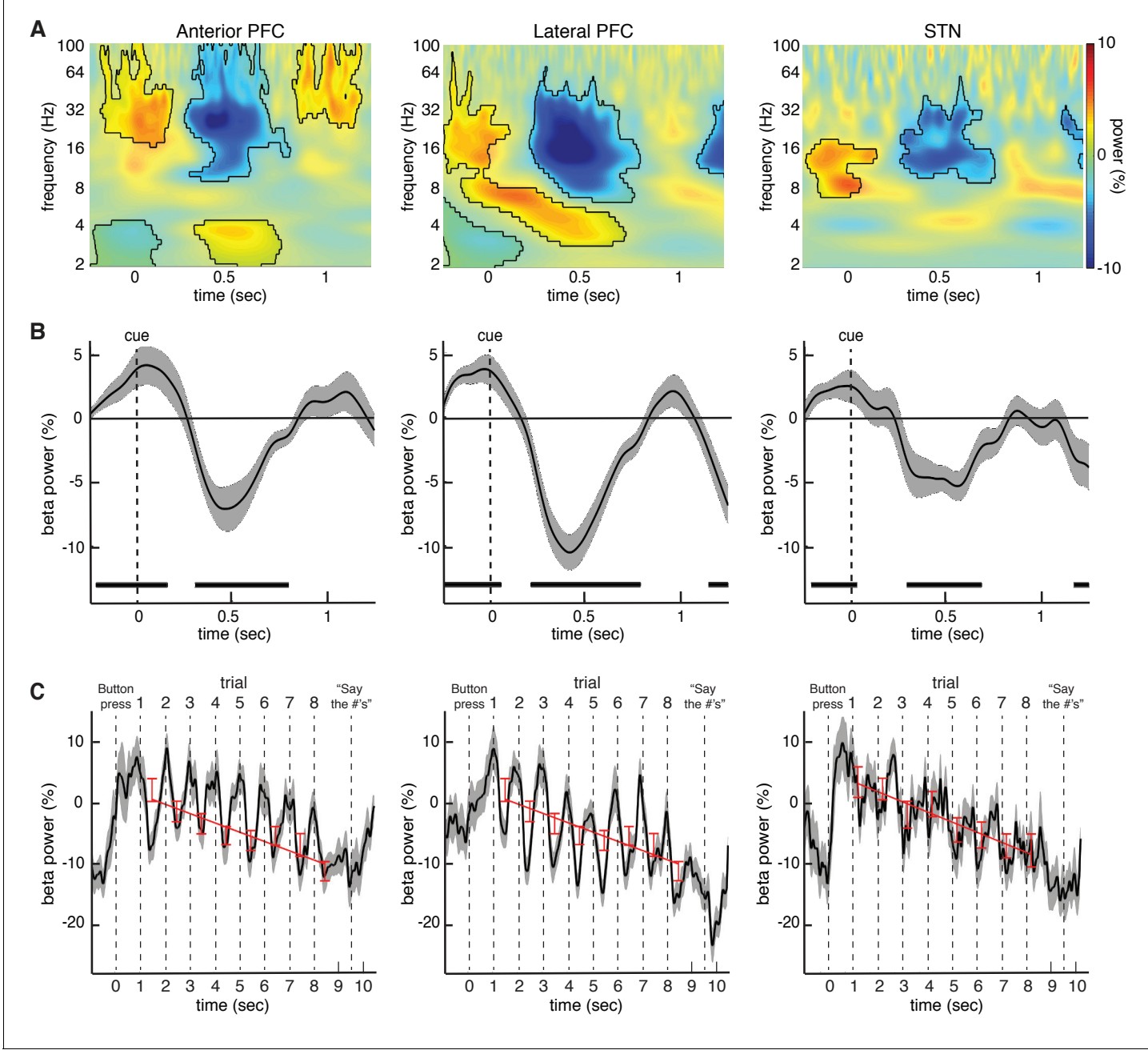

**Figure 2.** Changes in beta power during the task relative to baseline. (A) Normalized oscillatory power averaged across all anterior PFC electrodes (left), all lateral PFC electrodes (middle), and all STN LFP macroelectrodes (right), averaged across all trials (target and distractor trials combined; numbers are presented at t = 0; mask indicates time frequency regions exhibiting significant differences from baseline at p < 0.05, corrected for multiple comparisons, permutation test). All three recording sites showed a decrease in beta (15–30 Hz) power during the task. (B) Same as (A), but averaged across the entire beta band. Time points exhibiting a significant change from baseline (p < 0.05, corrected for multiple comparisons, permutation test) are denoted by black horizontal bar. (C) Progressive changes in beta power across all eight trials within a block averaged across all blocks. As in (A), the data were averaged over all anterior PFC electrodes (left), lateral PFC electrodes (middle), and STN macroelectrodes (right). Stimuli were presented at t = 1, t = 2, t = 3, etc. Red bars: Average beta power at every trial within a block. Best fit line from a linear regression of beta power against trial number is shown. In all three brain regions, there was a significant overall decrease in beta power over time during each block of 8 trials (*p*<0.05, repeated-measures ANOVA).

DOI: https://doi.org/10.7554/eLife.31007.005

The following figure supplement is available for figure 2:

**Figure supplement 1.** Changes in power during the task relative to baseline for all lateral PFC electrodes in three participants.

*Figure 2 continued on next page*

*Figure 2 continued*

DOI: https://doi.org/10.7554/eLife.31007.006

average power separately across all target and distractor trials during each experimental session. Across sessions, all three brain regions exhibited initial decreases in beta power following the presentation of both target and distractor. However, during the distractor trials, both the STN and lateral PFC demonstrated an early termination of the beta band decrease. This resulted in significantly higher beta band power relative to the target trials (p = 0.020 for STN and p = 0.025 for lateral PFC, permutation test; *Figure 3*; see Materials and methods; for changes in theta band power, see *Figure 3—figure supplement 1B*).

As with the changes in overall beta power during the task, we were also interested in whether the observed differences in beta power between target and distractor trials were specific to individual electrode contacts. We found that 13 out of the 24 experimental sessions with lateral PFC electrodes demonstrated a significant difference between target and distractor trials in at least one contact (p < 0.05, permutation test). Indeed, the observed effect averaged across the entire brain region was only mediated by a subset of electrodes. Within these 13 sessions, only 2.1 ± 0.2 of seven electrode contacts (29.7 ± 3.4%) showed a significant difference in beta power between target and distractor trials (p < 0.05, permutation test; *Figure 3—figure supplement 2*). Conversely, in the anterior PFC, only 6 out of 29 sessions demonstrated at least one contact exhibiting a significant difference in average beta power between target and distractor trials (p < 0.05, permutation test).

Although our data demonstrate that target and distractor stimuli involved differences in beta oscillatory power during each trial, we were interested in whether these differences were directly related to the behavioral performance on each trial. As such, we next examined the relatively few

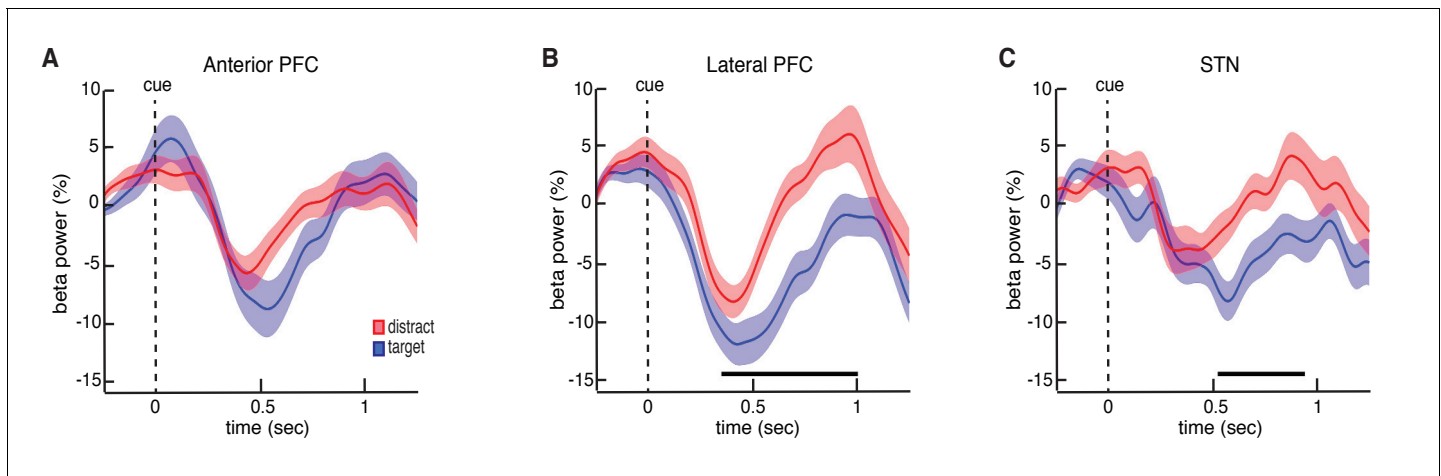

**Figure 3.** Trial type related differences in beta power. (A) Cue-aligned time evolving beta power changes averaged over all anterior PFC electrodes during the target and distractor conditions. (B) Same as (A) but for the lateral PFC. (C) Same as (A) but for the STN. Time points exhibiting a significant difference (p < 0.05, corrected for multiple comparisons, permutation test) are denoted by black horizontal bar. Both the lateral PFC and the STN showed significantly higher beta band power during the distractor condition.

DOI: https://doi.org/10.7554/eLife.31007.007

The following figure supplements are available for figure 3:

**Figure supplement 1.** Changes in theta power during the task relative to baseline.
DOI: https://doi.org/10.7554/eLife.31007.008
**Figure supplement 2.** Within-session analysis of trial type related differences in beta power.
DOI: https://doi.org/10.7554/eLife.31007.009
**Figure supplement 3.** Trial type related differences in beta power across time within a block.
DOI: https://doi.org/10.7554/eLife.31007.010
**Figure supplement 4.** Error-related differences in beta power.
DOI: https://doi.org/10.7554/eLife.31007.011

error trials in which participants failed to encode the target number. If participants failed to encode these target stimuli because they treated them as distractors, then physiologically the observed beta decreases should be more similar to the distractor rather than the target trials. In the lateral PFC electrodes, we found that the successfully encoded target trials exhibited significantly greater decreases in beta power than those target trials that participants failed to encode (p = 0.04, permutation test; *Figure 3—figure supplement 4*). Indeed, the changes in beta power during the target trials that were not encoded did not differ from the distractor trials, suggesting that the failure to encode those target trials were related to the same early rebounds in beta power that were present when participants intentionally chose not to encode the distractor numbers. We did not, however, find similar differences between the error trials and the successfully encoded target trials in the anterior PFC (p = 0.17, permutation test) or the STN (p = 0.20, permutation test, *Figure 3—figure supplement 4*). We also did not find significant differences between the erroneously encoded distractor trials and the correct distractor trials in any brain region (p > 0.05, permutation test, data not shown).

As the lateral PFC and the STN both demonstrated significant differences in oscillatory power between target and distractor trials, we were next interested in asking whether these conditions also exhibited significant differences in connectivity between the two regions. Within each participant, we first identified individual lateral and anterior PFC contacts exhibiting significant differences in power between target and distractor trials that mediated the overall changes in these brain regions (*Figure 3—figure supplement 2*). We focused our analysis of connectivity only on the lateral PFC electrodes that exhibited a significant difference in power within participants (p < 0.05, permutation test; 12 experimental sessions with simultaneously LFP recordings within the borders of the STN; see Materials and methods; an insufficient number of individual anterior PFC electrodes exhibited significant differences in power within participants, consistent with the lack of an overall effect of trial type in the anterior PFC). We examined connectivity by comparing the spectral coherence between the lateral PFC and STN to the coherence observed during a baseline period (see Materials and methods). As with the analysis of power, we first calculated the average coherence difference between target and distractor trials for the relevant STN-lateral PFC pairs within a session before testing for a significant trial-type related difference across all sessions. Distractor trials exhibited significantly higher coherence in the beta band between the lateral PFC and the STN than target trials

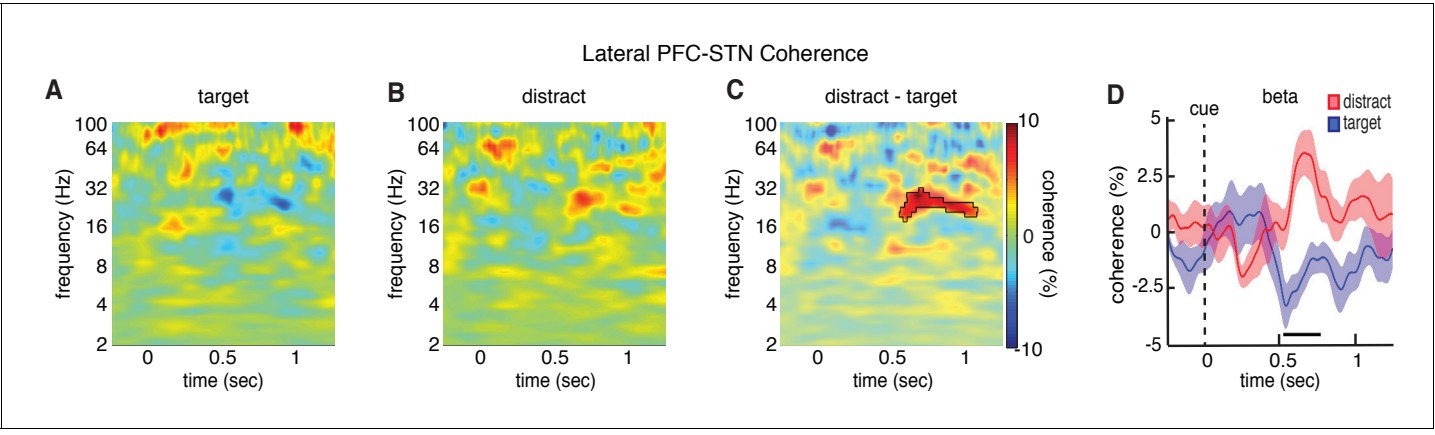

**Figure 4.** Differences in lateral PFC-STN coherence between target and distractor trials. Cue-aligned time evolving normalized coherence between the STN and the lateral PFC for the target (**A**) and distractor (**B**) conditions. (**C**) Normalized coherence difference between the distractor and target conditions (mask indicates time frequency regions exhibiting significant differences at p < 0.05, corrected for multiple comparisons, permutation test). (**D**) Cue-aligned time evolving beta coherence changes plotted separately for the target and distractor conditions. Time points exhibiting a significant difference (p < 0.05, corrected for multiple comparisons, permutation test) are denoted by black horizontal bar.

DOI: https://doi.org/10.7554/eLife.31007.012

The following figure supplement is available for figure 4:

**Figure supplement 1.** Trial-type-related differences in phase synchrony and power correlations Trial-type-related differences in phase synchrony and power correlations.

DOI: https://doi.org/10.7554/eLife.31007.013

(p = 0.015, permutation procedure; *Figure 4*), suggesting that the elevated levels of beta power in both brain regions during distractor trials were accompanied by elevated levels of beta band coherence. We subsequently tested whether the higher levels of coherence were due to increased synchrony of phase or increased correlation of power (*Cohen, 2014*). The increased coherence we observed during distractor trials was primarily driven by higher levels of phase synchrony (p = 0.025, permutation test; *Figure 4—figure supplement 1*). Distractor trials also had higher levels of correlation of beta power, but this difference did not survive multiple comparison testing.

Finally, we extracted spiking activity from microelectrode recordings in the STN while participants were engaged in the task. We identified spiking activity from 48 neuronal firing clusters across 21 STN recording sessions (35.9 ± 3.7 spikes/s (mean ± SEM) per cluster). On average, across all neurons, we found a significant decrease in overall spiking activity during each trial compared to baseline (p = 0.02, permutation test; see Materials and methods; *Figure 5A*). However, the responses of individual neuronal clusters were heterogenous. Some individual clusters demonstrated decreases in spiking activity during each trial, whereas others demonstrated increases (*Figure 5B*). Within individual recordings, we found that 47.9% (23 out of 48 clusters) of the recorded clusters demonstrated a significant change in firing rate (p < 0.05, permutation test) during the task (across all trials) when compared to baseline (*Figure 5C*). Of these neuronal clusters that showed a change in firing, most (65%, 15 out of 23 clusters) exhibited significant decreases in firing during the task, but some (35%, 8 out of 23 clusters) exhibited significant increases (average responses of both populations in *Figure 5D*). Relative to the dorsal and ventral borders of the STN, there were no significant differences in depth between the upward (mean depth 70.4 ± 7.8; see Materials and methods) and the downward firing clusters (mean depth 56.2 ± 5.4; p = 0.14, unpaired t-test).

We separately examined the neuronal clusters that individually demonstrated decreases in spiking activity during the task, and found that the average response of these clusters across an entire block exhibited similar transient dynamics during each trial to those observed in beta oscillatory power (*Figure 5E*). However, unlike the changes in beta oscillatory power, we did not find that there was a progressive change in overall spiking activity with subsequent trials across each block ($\rho = 0.13 \pm 0.20$, $t(14) = 0.61$, p = 0.55). Nevertheless, we examined the relationship between individual spike events and the ongoing LFP oscillations recorded from the STN macroelectrodes, and found that these neuronal clusters were significantly entrained to ongoing beta oscillations during the task (p = 0.02, permutation test; *Figure 5F*; see Materials and methods). Notably, the neuronal clusters that individually demonstrated increases in spiking activity during the task did not exhibit a significant interaction between LFP phase and spiking events (p = 0.16, permutation test; *Figure 5F*).

The observed changes in STN spiking activity suggest that indeed the STN modulates its spiking output during this non-motor task. However, as with the changes in beta oscillatory power, we were also interested in whether the spiking responses of these neurons were differentially modulated by target and distractor trials. Individual recordings accounting for approximately 20% of the recorded neuronal clusters exhibited significant differences in spiking activity between target and distractor trials (p < 0.05, permutation test; *Figure 5G*). However, we found no significant overall differences in spiking activity between target and distractor trials across either decreasing (p = 0.34, permutation test; *Figure 5H*) or increasing neurons (p = 0.36, permutation test). Moreover, although neuronal clusters exhibiting decreased spiking activity during the task were locked to the LFP beta oscillation, we found no significant differences in beta band spike phase locking between the target and distractor trials (p > 0.05, permutation test). This was the case when we examined the full 1000 ms window of all trials, as well as when we used smaller time windows (250 ms) centered at various points within the task (data not shown).

## Discussion

Our data demonstrate that the human STN modulates both oscillatory and spiking activity as participants make decisions regarding whether to attend to and remember individual items. Our data therefore build upon previous empiric findings suggesting that the basal ganglia may play a role in regulating working memory, and provide direct evidence that the STN exhibits changes in activity during a non-motor decision. Moreover, by simultaneously recording activity from the lateral PFC,

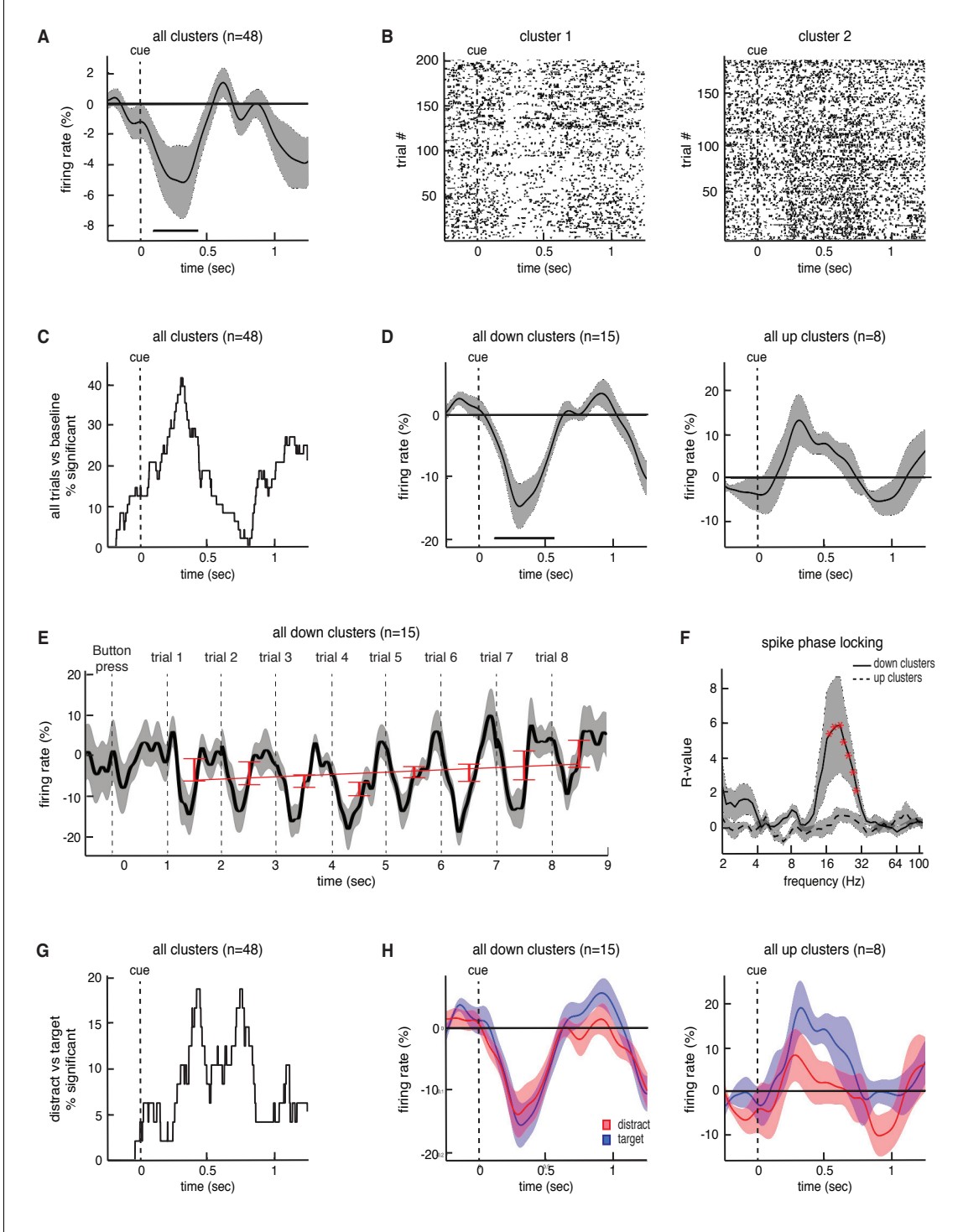

**Figure 5.** Task-related changes in spiking activity in STN neuronal clusters. (**A**) Average continuous-time firing rate for all 48 STN microelectrode recordings that showed spiking activity during the task (numbers presented at t = 0). Time points exhibiting a significant change from baseline (p < 0.05, corrected for multiple comparisons, permutation test) are denoted by black horizontal bar. (**B**) Raster plot for two neuronal clusters exhibiting changes in spiking activity during the task. (**C**) The percentage of recordings at each time point that showed a significant difference in firing during the task when comparing all trials to baseline. (**D**) Same as A but including only the 15 clusters that showed a significant decrease in firing during the task (left) and the eight clusters that showed a significant increase in firing during the task (right). (**E**) Progressive changes in firing rate across all eight trials within a block averaged across all blocks. The average across all 15 downward firing clusters is plotted. (**F**) The interaction between STN microelectrode spiking and macroelectrode LFP phases for each frequency, averaged across all down and up clusters with simultaneous LFP macro electrode

*Figure 5 continued on next page*

*Figure 5 continued*

recordings. Red asterisks indicate frequencies exhibiting significant spike-phase locking values, *R* (p < 0.05, corrected for multiple comparisons, permutation test). (**G**) The percentage of recordings at each time point that showed a significant difference in firing between the target and distractor trials. (**H**) Same as D but averaged separately for the target and distractor trial conditions.

DOI: https://doi.org/10.7554/eLife.31007.014

our data also demonstrate changes in oscillatory coherence between the PFC and STN, thereby tying cortical regions known to be involved in working memory with the STN.

In the novel working memory task used here, participants make internal decisions to attend to and encode numbers if they appear within a target shape, or to ignore them if they appear within a distractor shape. Importantly, both choices involve no immediate motor movements, enabling us to directly ask whether the STN participates in such non-motor decisions. As when inhibiting motor movements (*Zavala et al., 2015*), participants should optimally perform this task by immediately recognizing the indicator shape on each trial, and in the case of distractors, either ignore the presented item completely or prevent any processes that actively encode it. Although our data cannot distinguish these possibilities, the observed differences in neural activity suggest that participants are able to successfully make an internal distinction between the two trial types. Of note, this strategy would spare working memory capacity from storing distracting items that are unnecessary for correct completion of the block. It is possible, however, that the participants encode every number presented in each trial, along with the shape within which each number was presented. In this latter scenario, the participants would make a distinction between the target and distractor items only at the time of retrieval when vocalizing their response. Although we do not explicitly test their memory for the distractor items, it is less likely that participants are adopting this latter more challenging approach given how well they performed this task, and given the differences in neural activity observed at the time of presentation between target and distractor items.

The most robust changes we observed during our task in both the STN and in the lateral PFC were related to beta oscillatory activity. Changes in STN beta oscillatory power have been extensively studied in the context of movement, decreasing during the execution of a movement, and increasing when a movement is inhibited. The relation between beta oscillatory activity and movement is most evident during stop inhibition tasks when participants are asked to specifically begin or stop a motor movement (*Kühn et al., 2004*; *Ray et al., 2012*; *Alegre et al., 2013*). In the task described here, participants make a similar decision, albeit without moving. Remarkably, both the STN and the lateral PFC exhibit similar decreases in beta oscillatory power during this non-motor decision, with significantly earlier termination of these decreases when the presented item should be ignored.

The absence of movement here therefore allows us to specifically examine changes in STN activity during non-motor decisions and show that these changes are similar to those seen during movement inhibition. However, this also makes interpreting the timing of when these changes in beta power occur more difficult since there is no explicit behavioral measure of when each individual is internally processing the numbers and distractor signals, or when they are deciding to encode or ignore each stimulus. Nevertheless, we find that the timing of beta power differences between trial types is similar between our task and the tasks that required an inhibition of an actual movement, suggesting that the changes observed here are analogous to the previously identified stop signal (*Kühn et al., 2004*; *Ray et al., 2012*; *Alegre et al., 2013*). Moreover, in stop inhibition tasks, differences in beta power can still alter the behavior of the failed stop trials (in which a subject failed to properly inhibit an action) even when occurring after the erroneous response has already begun. These differences in beta power persist after that behavior has been initiated, and in the case of movement inhibition, this manifests itself as a suppression of the effort and duration of an erroneous response (*Ruiz et al., 2011*; *Cohen and van Gaal, 2014*). In the case of our current task then, increased STN beta activity may prevent the encoding of an item into working memory, or it may impair the maintenance of that item in memory. Although rebounds in beta power have also been observed following the execution of a movement, because these rebounds occur much later in the task than the timing observed here (*Alegre et al., 2013*), we think our data are more consistent with an internal stop signal rather than a post-action rebound in beta power.

Given the relative timing of the observed changes in beta power and the behavioral and physiological evidence that participants are able to successfully distinguish trial types, our data are therefore consistent with previous studies suggesting that beta band activity may in fact represent the brain's mechanism for globally suppressing all actions. The concept of global suppression has been well described in the context of movement, but more recently has been extended to include even non-motor processes such as working memory (*Wessel et al., 2016a*). Increases in STN beta oscillatory power have been observed when participants actively refrain from updating working memory (*Oswal et al., 2012*), and following the presentation of a surprise auditory stimulus that impaired the maintenance of items in working memory (*Wessel et al., 2016b*). Moreover, inhibiting movements can result in an unintentional inhibition of memory (*Chiu and Egner, 2015*), suggesting that the suppression of actions related to movement and to memory may be linked.

Indeed, the suggestion that beta band oscillatory activity could serve as a global suppression signal for both motor and non-motor functions throughout the brain finds support both in our observation that the STN and the lateral PFC exhibit similar decreases and rebounds in beta oscillatory power during this non-motor decision, and in previous studies of the PFC. In particular, beta oscillations in the lateral PFC decrease during the encoding phase of working memory (*Spitzer et al., 2010*; *Brookes et al., 2011*; *Heinrichs-Graham and Wilson, 2015*; *Kornblith et al., 2016*; *Lundqvist et al., 2016*; *Wiesman et al., 2016*), and exhibit similar changes as found in the STN when movements are inhibited (*Swann et al., 2009*, *2012*). In patients with Parkinson's disease, therefore, excessive beta oscillatory activity may not only impact movement, but could also play a role in the observed deficits in working memory due to the global nature of this activity (*Owen et al., 1997*; *Lewis et al., 2003*; *Chiaravalloti et al., 2014*; *Trujillo et al., 2015*; *Wiesman et al., 2016*). These effects may be related to the loss of dopaminergic innervation to the lateral PFC that result in increased beta oscillations (*Puig and Miller, 2012*; *Puig et al., 2014*; *Puig and Miller, 2015*). Importantly, here we demonstrate that the changes in beta oscillatory power are accompanied by significant changes in beta oscillatory coherence between the lateral PFC and the STN. Although the lack of intraoperative CT imaging limited our ability to determine precisely what areas of the lateral PFC were interacting with the STN, our data provide direct evidence linking these structures and their respective roles as participants make these non-motor decisions related to memory.

Interestingly, the progressive decreases in overall beta oscillatory power over successive trials observed here suggest a link with working memory load. Previous studies have demonstrated progressive increases in gamma band power in the PFC and hippocampus with working memory load (*van Vugt et al., 2010*; *Roux et al., 2012*), and in alpha band power in the parietal lobe (*Tuladhar et al., 2007*). These progressive changes are not consistent, however, as progressive decreases in oscillatory activity with working memory load have also been observed across several cortical regions (*Meltzer et al., 2007*), similar to our data. Of note, we observed such progressive decreases separately for both target and distractor trials, with no progression in the differences in beta oscillatory power between them. It is also possible, therefore, that these progressive decreases are not related to memory load, and instead emerge as each trial brings the participant closer to the forthcoming motor movement at the end of each list. Conversely, if these progressive decreases are indeed linked with memory load, then participants may be retaining some memory of the distractor items even though, during individual trials, their neural data suggest that they treat these items differently. Alternatively, the progressive decreases could be related only to increasing memory load of the target items, in which case the distractor trials may not add to the decreases, but also may not reverse them.

Ultimately, the STN exerts its effects on the rest of the basal ganglia through its spiking activity. Our data demonstrate that the firing rates of a significant fraction of individual neurons within the STN were also modulated during this working memory task that did not involve an explicit motor movement. Overall firing rates averaged across all neurons decreased during each trial, consistent with the suggestion that decreases in overall STN spiking activity are facilitatory. However, just as movement elicits variability in the spiking responses of individual neurons (*Zavala et al., 2017*), the decision to attend to and encode items was accompanied by heterogeneous spiking responses. Indeed, the observed overall decreases in spiking activity was mediated by only a subset of neurons. Interestingly, these neurons also exhibited significant phase locking to the ongoing beta oscillations, suggesting a link between spiking activity in the STN and LFP oscillations.

Despite the overall decreases in spiking activity, however, only a subset of neurons demonstrated a significant difference in spiking activity between target and distractor trials. Moreover, we did not find evidence that there was a significant difference in spike phase locking between the two conditions. Hence, while our oscillatory data would implicate the basal ganglia at large in such non-motor decisions that distinguish target and distractor items, the spiking data would suggest that only a subset of STN neurons are involved in this process, and that these neurons do not alter the extent to which they are locked to the ongoing beta oscillations accordingly.

Finally, our data demonstrate that the changes in cortical beta oscillatory power were also accompanied by changes in cortical theta oscillatory power. Unlike beta oscillations, the precise roles played by theta oscillations in the cortex and STN are less clear. The elevated levels of cortical theta oscillatory power during the distractor trials are consistent with the role cortical theta oscillations may play during conflict related response inhibition (*Cavanagh et al., 2011*, *2012*). Given our previous work demonstrating increases in STN theta power and cortico-STN theta coherence during conflict (*Zavala et al., 2014*, *2016b*, *2016a*), however, we were surprised to find no trial-type related differences in theta power in the STN during this task. One possibility, however, is that STN theta increases are specific to conflict induced slowing of response times rather than complete response inhibition. Indeed, previous studies have demonstrated that motor response inhibition as tested using Go-NoGo paradigms fails to elicit differences in theta oscillatory activity despite showing robust changes in the beta band power similar to those reported here (*Kühn et al., 2004*; *Ray et al., 2012*; *Alegre et al., 2013*).

Together, our data demonstrate that many of the circuit mechanisms of the basal ganglia that have been previously described in the context of movement are used in an analogous fashion during non-motor cognitive decisions. Although our data were collected from patients with Parkinson's disease in an intraoperative environment, we primarily contrast beta band activity when participants were making decisions between different trial types, suggesting that these differences may exist or even be enhanced (*Oswal et al., 2012*) in the healthy brain. Here, we were specifically interested in the non-motor decision to attend to and encode items into working memory. Our data raise the possibility, however, that similar neural mechanisms and interactions between the basal ganglia and the cortex could be used for any non-motor decision that involves proceeding with or aborting a cognitive process.

## Materials and methods

### Intraoperative task and recordings during deep brain stimulation surgery

We captured intraoperative recordings in 18 participants undergoing deep brain stimulation (DBS) surgery of the subthalamic nucleus (STN) for Parkinson's disease. The study was conducted in accordance with an NIH IRB approved protocol, and all participants gave their written informed consent to take part in the study. Participants received no financial compensation for their participation. Parkinson's medications were stopped on the night before surgery (12 hr preoperatively). We captured recordings while participants were alert, at rest, and in an OFF state while in the operating room.

As per routine DBS surgery, we used microelectrode recordings to identify the STN based on firing rate and pattern. We simultaneously advanced three electrodes, separately spaced 2 mm apart, during each recording session (placed along a central, 2 mm lateral, and 2 mm anterior trajectory; *Figure 1C*). Each targeting electrode consisted of a microelectrode contact and a macroelectrode contact positioned 3 mm dorsal to the microelectrode tip. Macroelectrode were within the STN if the corresponding microelectrode contact was greater than 3 mm ventral to the dorsal border of the STN (identified by increased spiking activity and background noise relative to the more dorsal zona incerta and thalamus). We restricted all analyses only to signals captured from electrode contacts positioned within the STN. Raw signals were sampled at 1.5024 and 24.0345 kHz from macro and microelectrode contacts, respectively, and stored using a MicroGuide Pro data acquisition system (Alpha Omega Co., Alpharetta, GA). The raw electrophysiology data and relevant code are available on Dryad (*Zavala et al., 2018*).

During the operative procedure, we acquired simultaneous intracranial EEG (iEEG) recordings from two subdural strip electrodes temporarily placed through the DBS burrhole PMT Corporation,

Chanhassen, MN; *Figure 1C*). We placed a six-contact anterior prefrontal cortex (PFC) strip electrode consisting of a single row of six platinum contacts (2.3 mm exposed diameter with 1 cm inter-contact spacing) in a direct anterior direction from the burr hole. We also placed an eight-contact lateral PFC strip electrode in a direction that was angled approximately 60 degrees lateral to the direction of the anterior PFC strip electrodes. We confirmed contact localization using intraoperative X-ray (*Figure 1—figure supplement 1*). The subdural strip electrodes were removed after completion of the behavioral task on each side.

## Behavioral task

Participants performed a novel working memory task (*Figure 1A*) in the intra-operative environment. The task involved 30 blocks. During each block, we sequentially presented participants with eight randomly chosen single-digit numbers, and instructed them to attend to and encode four of the numbers (targets) and ignore the remaining four numbers (distractors). Each number was presented within either a square or an octagon. At the beginning of each block, we pseudo-randomly chose and displayed in the center of the screen either an image of the square or the octagon to indicate the target shape for the upcoming block. The participants indicated that they understood the target shape by pressing a large handheld button. After the button press, we displayed a blank screen for $1000 \pm 100$ ms. We then sequentially displayed each number for 500 ms, followed by a blank screen for $500 \pm 100$ ms. In this manner, each block consisted of four target trials and four distractor trials that were randomly interleaved. No numbers were repeated. During the presentation of the numbers, we instructed the participants not to make any responses and to explicitly limit all movements if possible.

Following the presentation of the final number in each block, a blank screen was presented on the screen for one second. We then prompted the participants to vocally retrieve the four target numbers by displaying the words 'Say the numbers' on the screen. We manually recorded their responses by typing it into the computer. If the participant correctly vocalized all four target numbers (in any order), and said no other numbers, we presented a green smiley face and the word 'CORRECT' in the center of the screen. If the participant did not correctly vocalize the four target numbers, we presented a red sad face and the word 'WRONG.' In the rare case when the participant vocalized only the four distractor numbers (and none of the target numbers), we presented a yellow neutral face and the phrase, 'Wrong Symbol.' In these cases, the participant had performed the task correctly, but had encoded and retrieved numbers presented within the wrong shape. Feedback in this case was intended to encourage participants to pay more attention to the target shape presented at the beginning of each block. For subsequent analysis, we included all trials from the correct blocks and from blocks in which the participant attended to the wrong shape because in both cases the participant had attended to just one shape and ignored the other.

Most participants completed 30 blocks of 8 trials each (four target and four distractor trials) during each experimental session. Despite the challenging setting, most participants had were able to successfully remember all four numbers. During three sessions, however, the participant failed to correctly encode and retrieve the target numbers during several of the initial blocks but was willing to continue performing the task. In these three cases, we added additional blocks at the end of the experimental session in order to increase the total number of correct blocks included in the analysis. Participants correctly completed $22.8 \pm 0.5$ blocks, corresponding to $91.0 \pm 2.1$ target trials per participant (and an equal number of distractor trials), that were included in our main analysis. During incorrect blocks, participants made two types of errors. They either failed to vocalize a target number, or they erroneously vocalized a distractor number. For our analysis of error-related activity, we only included sessions in which participants made these errors in at least five blocks. Errors in which participants failed to encode target items met this inclusion criteria in 24 sessions, resulting in $14.8 \pm 2.8$ error trials (meaning trials in which the target number being displayed was not subsequently remembered) per session. Errors in which participants erroneously vocalized distractor items met this inclusion criteria in 18 sessions, resulting in $8.6 \pm 1.9$ error trials (meaning trials in which the distractor number being displayed was subsequently recalled) per session.

On the day before surgery, participants practiced a complete 30-block session of the task in order to familiarize themselves with the task. Prior to the practice session, we explained the three possible types of feedback the participants could potentially receive at the end of each block and emphasized the importance of not moving during the time period when the numbers were displayed.

During the operative procedure, most participants performed one session while we captured recordings from the left STN and a second session while we recorded from the right STN. Four of the 18 participants did not complete a second session on the opposite side because of fatigue. Thus, there were 32 total intraoperative STN recordings included in the analysis. In two of these 32 experimental sessions, the LFP macro electrodes were not located within the STN. Furthermore, in three of the 32 experimental sessions, we did not implant any iEEG strip electrodes, and in five additional sessions we only implanted the anterior PFC strip electrodes without the additional lateral PFC strip electrodes. We collected the data during an 18-month period between Nov 2014 and April 2016. Every patient who was willing and able to participate in the task was included in the analysis.

## LFP and iEEG power

We performed all analyses using custom MATLAB scripts (Mathworks, Natick, MA; Raw data and relevant analysis code are available on Dryad [*Zavala et al., 2018*]). We extracted local field potential (LFP) activity from each macroelectrode and iEEG activity from each subdural contact. We bandpass filtered both signals between 1 and 500 Hz, notch filtering at 60 Hz, and downsampled the data to 1 kHz. We referenced macroelectrode signals to the common average of all simultaneously recorded macroelectrode contacts to yield three referenced monopolar LFP channels per session. We referenced the iEEG signals by subtracting the signals of adjacent electrodes. We henceforth refer to these bipolar channels as electrode contacts. Prior to any subsequent analysis, we manually discarded all trials exhibiting a clear artifact in the LFP or iEEG trace.

In order to obtain magnitude and instantaneous phase information in the frequency domain, we convolved the LFP signals captured from the STN and iEEG signals captured from the subdural contacts from each trial with complex valued Morlet wavelets (wave number 6). We used 47 logarithmically spaced (8 scales/octave) wavelets between 2 and 107 Hz and convolved each wavelet with 1500 ms of LFP data from each trial (250 ms before to 1250 ms following number presentation). We used a 1000 ms buffer on both sides of the clipped data to eliminate edge effects. We squared the magnitude of the continuous-time wavelet transform to generate a continuous measure of instantaneous power for each frequency. We determined the percentage change in power (normalized power) for each channel and frequency by comparing the continuous measure of power to the mean power recorded from that channel during a baseline period. We defined the baseline period as the 500 ms preceding the presentation of each number.

For each STN recording, we averaged the normalized power from macroelectrodes that were within the STN. Because we did not have access to intraoperative computed tomography (CT) imaging, we were unable to use patient specific landmarks to accurately localize individual iEEG contacts. Thus, we defined the activity for each cortical brain region as the averaged normalized power from all bipolar channels recorded from that strip. This procedure therefore resulted in a single spectrogram that we broadly assigned to each of the two cortical brain regions (anterior and lateral PFC) during each trial.

## Statistical analysis

To assess overall changes in spectral power regardless of trial type during the task compared to baseline, we performed a random-effects statistical analysis in each brain region across experimental sessions. We treated each experimental session as an independent event since in the cases when a participant completed two experimental sessions, they did so while we recorded from the left and right hemispheres separately. We first calculated the trial-averaged normalized power for each region, yielding an average normalized power spectrogram at each time point and frequency for each session. Our null hypothesis was that across experimental sessions, the average normalized power at each time-frequency point would be zero (no change from baseline). We tested this hypothesis using a non-parametric permutation procedure in which the across trial average for each experimental session is the unit of observation (*Maris and Oostenveld, 2007*).

In each region, we computed the true mean difference in spectral power between task and baseline in each session for every time point and frequency. This generates a distribution of spectrograms (one for each session) and a resulting average spectrogram across experimental sessions. We then generated a surrogate distribution of differences by permuting the data 200 times to generate an empiric distribution of possible mean spectrograms that are all equally probable under the null

hypothesis. Each of the 200 surrogate mean spectrograms was generated by randomly either subtracting the baseline from the task data or subtracting the task data from the baseline in each session, and then recomputing the mean difference in the spectrogram across sessions. For every time-frequency point, we then compared the true mean across-session power to the mean and standard deviation of the corresponding point in the empiric distribution to generate a p-value. This p-value represents the likelihood that the true mean change from baseline for each time-frequency point represents a departure from the null hypothesis. However, this p-value for each time-frequency point does not take into account the multiple comparisons that are made across all time points and frequencies.

To correct for multiple comparisons across all time points and frequencies, we used a cluster correction method based on exceedance mass testing (*Maris and Oostenveld, 2007*). This method assumes that a true effect at any time-frequency point is likely to be observed across multiple time points and frequencies. We defined time-frequency clusters by thresholding the across-session *p*-values derived from the statistical analysis described above. Any contiguous time-freqency points with a p-value less than $0.05$ were included in each cluster. For each identified cluster, we defined a cluster statistic to be the sum of the *z*-scores, derived from the p-value using a normal cumulative distribution function, for all time-frequency points within that cluster.

We calculated clusters using the true data, and for each of the 200 permutations. We used the maximum cluster statistic of each permutation to create an empiric distribution for significance testing. We determined whether a true cluster test statistic was significant by comparing it to the empiric distribution of maximum cluster test statistics. In this manner, significant clusters can arise from large changes in power that extend over a small number of frequencies or over a small time period, or from smaller differences that involve a larger number of time-frequency points. We considered cluster test statistics with $p < 0.05$ to be significant and corrected for multiple comparisons.

We also tested whether there were any differences in response to the task that depended on the hemisphere (left or right) from where the data were recorded. For all of the participants that performed the task twice, we calculated the mean difference between the response on the left and right hemispheres. To test whether the difference we observed was significantly different from zero, we compared the true mean difference to the distribution of 200 surrogate differences calculated by randomly permuting the hemisphere labels (left or right) of each experimental session's across-trial average prior to averaging across sessions. We tested for significance and corrected for multiple comparisons using the same cluster based procedure described above.

We also tested for significant changes in power within two specific frequency bands of interest (theta, 2–8 Hz; beta, 15–30 Hz) based on prior evidence suggesting these frequency bands are involved in conflict and response inhibition (*Cavanagh et al., 2011*, *2012*; *Cohen and Cavanagh, 2011*; *Brittain et al., 2012*; *Zavala et al., 2013*; *2014*, *2017*; *Kühn et al., 2004*; *Alegre et al., 2013*). We used the same permutation procedure described above, but first averaged spectral power across the frequency band of interest prior to calculating any true or surrogate means. In this case, clusters were based on contiguous time points exhibiting significant differences across sessions.

To identify individual electrode contacts within individual experimental sessions that demonstrated a significant change in power, we used the same statistical analysis and permutation procedure described above. In this case, however, the unit of observation for our statistical test was the individual trial. Hence, for each electrode contact, we calculated the difference in spectral power between each trial and baseline, and computed the mean across all trials. We then compared that true mean spectral power to a distribution of 200 surrogate mean spectrograms. Each of the 200 surrogates was generated by randomly either subtracting the baseline from spectral power during the trial or subtracting the spectral power during the trial from the baseline in each trial, and then recomputing the mean difference in the spectrogram across trials.

To assess differences in spectral power between conditions (target and distractor) across experimental sessions, we used an identical permutation procedure. In this case, however, we calculated the average normalized spectral power separately for target and distractor trials in each experimental session. We then calculated the average true mean difference between target and distractor conditions across sessions. We again performed a random-effects statistical analysis, with our null hypothesis in this case being that across sessions, there was no difference in normalized power between target and distractor trials. Thus, to test whether the difference we observed was

significantly different from zero, we compared the true mean difference to the distribution of 200 surrogate differences calculated by randomly permuting the condition labels (target or distractor) of each experimental session's across-trial average prior to averaging across sessions. We tested for significance and corrected for multiple comparisons using the same cluster based procedure described above. We used a similar procedure to compare the correct target trials to the incorrect target trials as well as the correct distractor trials to the incorrect distractor trials.

To determine whether any individual electrode contacts exhibited a significant difference in spectral power between target and distractor trials, we performed the same statistical analysis and permutation procedure. In this case, for each electrode contact, we compared the mean spectral power difference between target and distractor trials to a surrogate distribution generated by permuting the trial labels (target or distractor) within that individual session. We identified 13 of the 24 recordings sessions that had lateral electrodes that showed a significantly higher power during the distractor trials for at least one contact. Twelve of these recording sessions had simultaneous LFP recordings within the borders of the STN. For the anterior PFC electrodes, 6 of the 29 recording sessions contained electrodes with a significant difference in power, but only 4 of these sessions had simultaneous LFP recordings within the borders of the STN. For this reason, we focused our subsequent analyses only on the 12 lateral PFC electrode recordings that had significant power differences and simultaneous LFP recordings within the borders of the STN.

## STN-cortical coherence

To estimate the time-varying coherence between the STN and the lateral PFC, we calculated the cross-spectrum between every pair of contacts at each time-frequency point by multiplying the complex value extracted from the LFP signal at each time-frequency by the conjugate of the complex value extracted from the iEEG signal. We similarly calculated each signal's auto-spectrum at every time and frequency. To generate a continuous-time estimate of coherence, we used the absolute value of the average cross-spectrum divided by the product of each signal's average autospectra over 250 ms sliding windows (step size 1 ms) (*Lachaux et al., 2002*). This results in a time-varying estimate of coherence for every time-frequency point, for every trial, and for every pair of electrodes. We determined the percentage change in coherence (normalized coherence) for each channel pair and time-frequency point by comparing the continuous measure of coherence to the mean coherence recorded from that channel pair during a baseline period (500 ms preceding each number). In order to compare differences in normalized coherence between target and distractor trials across participants, we used the same across-session permutation procedure described above.

Because changes in coherence can be due to increased phase synchrony or increased correlations of power (*Cohen, 2014*), we separately examined these two measures to further characterize the relation between the STN and the lateral PFC. We calculated phase synchrony using an identical method to our measure of spectral coherence. In this case, however, we extracted the angle of the cross-spectrum at each time point, which represents the phase difference between the LFP and iEEG signal, rather than magnitude. To compare differences in phase synchrony between target and distractor trials across sessions, we used the same permutation procedure described above.

To calculate the correlations in beta power between the STN and the lateral PFC, we first smoothed the time-frequency power data across time by convolving each trial's power time series with a 250 ms window separately for each frequency. At each time point, we calculated the Spearman correlation coefficient between the STN LFP power and the lateral PFC iEEG power across trials, resulting in a time-by-frequency spectrogram of correlations for each lateral PFC-STN electrode pair. In order to compare differences in power correlations between target and distractor trials across sessions, we used the same permutation procedure described above.

## Spiking activity

We extracted spiking activity by bandpass filtering microelectrode recordings between 0.3 and 3 kHz and resampling the filtered signals at 24 kHz. Using a spike-sorting software package (Plexon Offline Sorter, Inc., Dallas, TX), we identified spike waveforms by manually setting a negative or positive voltage threshold depending on the direction of the voltage deflection. Given the difficulty of isolating single-units in the STN (*Weinberger et al., 2006*; *Sharott et al., 2014*), it is possible that some of the units we recorded reflected the activity from more than one neuron. We will therefore

subsequently refer to each individual microelectrode recording as a neuronal cluster. We defined the location of each cluster as a percentage reflecting the relative depth between the dorsal and ventral border of the STN as defined by the microelectrode recordings.

We extracted 1500 ms of spiking data from each trial for each microelectrode. We excluded all trials with an average firing rate greater or less than 10 standard deviations from the average firing rate across all trials. We then calculated the continuous-time firing rates for each recording by smoothing the spike train from each trial (1 ms bins) with a Gaussian kernel (standard deviation 50 ms). To generate a normalized firing rate, we compared continuous time firing rates for each trial to the mean and standard deviation of the firing rates during the 500 ms baseline period and then averaged across trials.

To determine whether there were any overall changes in firing activity across all sessions, we used the same permutation procedure as described above, comparing the true mean continuous firing rate across all trials (both target and distractor trials) in each session to baseline and treating the session as the unit of observation. Similarly, we determined whether each individual cluster exhibited a significant change in firing during the task using the same permutation procedure, in this case comparing the differences in firing rate between trials and baseline across all trials within a given cluster and treating the trial as the unit of observation. We subsequently calculated for each time point the percentage of all recordings that showed a significant difference from baseline across all trials. Finally, to determine whether a given cluster exhibited a significant difference between the target and distractor conditions, we calculated the mean difference in firing at every time point between target and distractor trials. We then used the same within-session permutation procedure described above to test for significance. We subsequently calculated for each time point the percentage of all recordings that showed a significant difference between target and distractor trials.

## Spike-phase interactions

In order to analyze the relationship between neuronal firing and ongoing LFP phase oscillations, we calculated spike-phase interactions for each neuronal cluster (*Zavala et al., 2017*). Briefly, we tabulated the instantaneous phase of the macroelectrode LFP signal during all spiking events captured on the associated microelectrode during each trial. We then calculated the normalized mean vector length, ($r$), across all trials during the first 1000 ms following each trials stimulus onset. We normalized the resulting spike-phase locking values by permuting phase information across trials. Whereas in the true case, spike times from a given trial were assigned an instantaneous phase from that same time point in the same trial, in each permuted case we assigned to each spike time an instantaneous phase from the same time point in a different trial drawn at random from the pool of other trials. We permuted phase information 200 times resulting in a distribution of 200 surrogate $r$ values for each cluster. We compared the true $r$ value with the mean and standard deviation of the distribution of permuted $r$ values to generate a normalized spike-phase locking value for each neuronal cluster ($R$).

To determine if individual frequencies exhibited significantly non-zero $R$ values across all neuronal clusters, we used the same permutation procedure described above to correct for multiple comparisons. We first calculated the mean normalized spike-phase locking value, $R$, across all neuronal clusters. We then compared this true mean value to a distribution of 200 surrogate mean $R$ values. Each of the 200 surrogates was generated by randomly either subtracting zero from a neuronal cluster's $R$ value or subtracting the cluster's $R$ value from the from zero in each cluser. We then recomputed the mean $R$ value across clusters for each surrogate, and in each frequency, assigned a $p$ value based on the comparison between the true mean $R$ and the surrogate distribution. To correct for multiple comparisons across frequencies, we used the same cluster correction method based on exceedance mass testing (*Maris and Oostenveld, 2007*). In this case, clusters of significant frequencies were based on contiguous frequency points exhibiting significantly non-zero $R$ values across neuronal clusters. Because only 10 of the 15 downward firing clusters had simultaneously recorded macroelectrodes within the borders of the STN, only these 10 clusters were included in the analysis. All 8 upward firing clusters had simultaneously recorded LFP electrodes within the STN, so all 8 clusters were used.

We subsequently repeated the analysis separately for the target and distractor trials. The trial scrambling method used to generate the $R$ values allowed us to normalize spike-phase locking values observed for a given trial type by the probability of observing spike-phase locking by chance given the distribution of phases and spike times observed during that trial type. In order to compare

differences in spike-phase-locking between target and distractor trials across clusters, we used the same permutation procedure described above.

## Supplementary information

### Electrode locations

Mean coordinates of the central microelectrode STN recording sites during the behavioral task, referenced to the mid-commissural point, were x = 12.0 ± 0.3, y = 5.6 ± 0.8, and z = 4.4 ± 0.3 for left electrode recordings, and x = −11.7 ± 0.3, y = 5.9 ± 0.9, and z = 5.0 ± 0.4 for right electrode recordings. These coordinates correspond to left and right STN on the Schaltenbrand-Wahren brain atlas.

In order to estimate locations of the subdural contacts, we co-registered the post-op CT with the pre-op T1 weighted MRI using the publically available software packages AFNI (http://adni.nimh.nih.gov) (*Cox, 1996*; *Saad and Reynolds, 2012*). We skull-stripped the pre-operative T1 weighted images. We performed an affine transform using AFNI's 'align-epi-anat.py' with the LPC cost function. We deobliqued and inverted the intensity of the MRI, and center aligned the CT to the MRI. If the registration was unsuccessful, we reprocessed the CT to remove air voxels, and attempted alignment again. On the co-registered image, we identified the location of the burrhole on the pre-operative MRI (*Figure 1—figure supplement 1B*). We estimated contact locations by measuring every 1 cm directly anterior (for the six contact anterior PFC strips) and 60 degrees antero-laterally (for the eight contact lateral PFC strips) from the burrhole. As this procedure does not provide accurate electrode contact locations, we only used these estimates to confirm that subdural strip electrode locations approximately corresponded to the anterior and lateral PFC. In order to compare electrode locations across subjects, we used surface-based registration to plot each electrode at its corresponding mesh location on the standard N27 'Colin' brain (*Holmes et al., 1998*), then, for visualization purposes, projected these electrodes to an approximated dural surface and imposed each strip's original geometric alignment. In this manner, electrodes appeared in analogous anatomical locations while laying visibly above the pial surface and in the correct geometric layout. Electrode projection estimates were unavailable for four of the 18 participants because one of the participants was not implanted with iEEG electrodes, and three additional participants had CT scans that did not include the top of the skull. Thus, the location of the burrhole could not be determined for these three participants.

### Additional analysis of changes in oscillatory power

The theta (2–8 Hz) time series for the anterior PFC, the lateral PFC, and the STN are shown in *Figure 3—figure supplement 1*. These data are the same as those shown in *Figure 2A*, however, here the analyses are restricted to the theta band. We found significant increases in theta band (2–8 Hz) power in the anterior and lateral PFC (p < 0.005 for anterior and lateral PFC, permutation test) during the task. Theta band power was significantly higher during the distractor trials in the anterior and lateral PFC (p = 0.03 for anterior PFC and p = 0.04 for lateral PFC, permutation test; *Figure 3—figure supplement 1B*) but not in the STN.

We examined the progressive decrease in beta oscillatory power separately for target and distractor trials (*Figure 3—figure supplement 3*). Both target and distractor trials demonstrated a significant decrease in beta power throughout each block (anterior PFC target stimuli $\rho = -0.27 \pm 0.1$, $t(27) = 2.7$, p = 0.01; lateral PFC targets $\rho = -0.34 \pm 0.1$, $t(23) = 2.96$, p = 0.01; STN target stimuli $\rho = -.29 \pm 0.1$, $t(28) = 2.55$, p = 0.02; anterior PFC distractor stimuli $\rho = -0.36 \pm 0.1$, $t(27) = 3.5$, p = 0.002; lateral PFC distract stimuli $\rho = -0.23 \pm 0.1$, $t(23) = 1.88$, p = 0.07; STN distractor stimuli $\rho = -0.37 \pm 0.12$, $t(28) = 3.03$, p = 0.005). In the lateral PFC and the STN, the two regions that showed a significant overall difference in beta power between target and distractor trials (*Figure 3*), we observed an effect of trial time (1 st trial, 2nd trial, 3rd trial, 4th trial) and trial type (target vs distractor), but we did not observe an interaction between the two (repeated-measures ANOVA, trial numbers X trial type, lateral PFC: trial type $F = 8.69$, $df = 1$, p = 0.007; time, $F = 10.1$, $df = 3$, p < 0.001; interaction $F = 0.39$, $df = 3$, p = 0.76; STN: trial type $F = 5.31$, $df = 1$, p = 0.03; time, $F = 6.12$, $df = 3$, p < 0.001; interaction $F = 1.01$, $df = 3$, p = 0.39). In contrast, the anterior PFC did not show an effect of trial type, but did show an effect of trial time and an interaction between trial

type and trial time (trial type $F = 2.68$, $df = 1$, p = 0.11; time, $F = 10.5$, $df = 3$, p < 0.001; interaction $F = 3.12$, $df = 3$, p = 0.03).

We examined the changes in beta power during the target trials that were erroneously not encoded. In the lateral PFC electrodes, we found that the successfully encoded target trials exhibited significantly greater decreases in beta power than those target trials that participants failed to encode (p = 0.04, permutation test; *Figure 3—figure supplement 4*). We did not find similar differences between the error trials and the successfully encoded target trials in the anterior PFC or the STN. We also did not find significant differences between the erroneously encoded distractor trials and the correct distractor trials in any brain region (p > 0.05, permutation test, data not shown).

The individual contact power results are shown for every lateral iEEG recording for one session from one participant (*Figure 3—figure supplement 2a*). In this session, four of the seven bipolar recordings showed significantly higher power during the distractor condition relative to the target condition. These four electrodes were the ones subsequently included in the analysis of STN-lateral cortex coherence for this participant. The estimated location of these electrodes on that participants brain are shown in *Figure 3—figure supplement 2a*. The projected locations of all lateral PFC electrodes that showed a significant difference in power between target and distractor trials are plotted on a standard brain in *Figure 3—figure supplement 2b*.

## Acknowledgements

We thank Codrin Lungu and Nora Vanegas-Arroyave for their help in intra-operative localization of the subthalamic nucleus and for the medical care and management of the participants in this study. We thank Michael Trotta for his help localizing electrodes. This work was supported by the Intramural Research Program at the National Institutes of Health. We are indebted to all patients who have selflessly volunteered their time to participate in this study.

## Additional information

### Funding

| Funder | Grant reference number | Author |
| --- | --- | --- |
| National Institute of Neurological Disorders and Stroke | Intramural Research Program | Kareem A Zaghloul |

The funders had no role in study design, data collection and interpretation, or the decision to submit the work for publication.

### Author contributions

Baltazar A Zavala, Conceptualization, Data curation, Formal analysis, Investigation, Methodology, Writing—original draft; Anthony I Jang, Data curation, Formal analysis, Writing—review and editing; Kareem A Zaghloul, Conceptualization, Data curation, Supervision, Funding acquisition, Methodology, Project administration, Writing—review and editing

### Author ORCIDs

Kareem A Zaghloul (ID) http://orcid.org/0000-0001-8575-3578

### Ethics

Human subjects: The study was conducted in accordance with an NIH IRB approved protocol (11-N-0211), and all participants gave their written informed consent to take part in the study. Participants received no financial compensation for their participation.

### Decision letter and Author response

Decision letter https://doi.org/10.7554/eLife.31007.019
Author response https://doi.org/10.7554/eLife.31007.020

## Additional files

### Supplementary files
• Transparent reporting form
DOI: https://doi.org/10.7554/eLife.31007.015

### Major datasets
The following dataset was generated:

| Author(s) | Year | Dataset title | Dataset URL | Database, license, and accessibility information |
|---|---|---|---|---|
| Zavala BA, Jang A, Zaghloul KA | 2018 | Data from: Human subthalamic nucleus activity during non-motor decision making | http://dx.doi.org/10.5061/dryad.9t740 | Available at Dryad Digital Repository under a CC0 Public Domain Dedication |

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
