## [Decision Letter]

Thank you for submitting your article "Human subthalamic nucleus activity during non-motor decision making" for consideration by *eLife*. Your article has been reviewed by two peer reviewers, and the evaluation has been overseen by David Badre as the Reviewing Editor, and Michael Frank as the Senior Editor. The following individual involved in review of your submission has agreed to reveal his identity: James Cavanagh (Reviewer #3).

The reviewers have discussed the reviews with one another and the Reviewing Editor has drafted this decision to help you prepare a revised submission.

Summary:

This study investigates the cortical and STN oscillatory dynamics during decision about working memory. LFP from prefrontal cortex and subthalamic nucleus (STN) along with spiking data from the STN are reported from human patients on the operating table performing a working memory task in which targets are encoded and not distractors. Beta power starts to decrease each time a letter is encoded, but if it's a distractor rather than a target, this quickly 'bounces' back up. Further, these effects are accompanied by coherence and spiking in the STN. These results provide novel evidence that the cortio-basal ganglia circuits may be involved in stopping cognitive actions akin to how they do motor ones.

Essential revisions:

1) In the Results and the Statistical analysis section, it is unclear whether data are being pooled across subjects (for a 'fixed effects' analysis) or analyzed separately for each subject then carried up to the group level to do statistics across subjects ('random effects' analysis). This has implications for the conclusions – if the former, then it is not possible to generalize beyond the population studied in the paper and this should be explicitly noted in the Results and Discussion section. Further, the reviewers agreed that in the absence of proper random effects analysis, analysis within each subject should be reported, along with the prevalence of the effect across individuals.

2) The cortical electrode locations were unclear and difficult to ascertain from the general labels, like "anterior PFC". The example participant reconstruction provided in the supplement does little to clarify. While Figure 1—figure supplement 1 has realistic/detailed reconstructions of electrode position, this is just for one subject. More detailed anatomical information about the position of the cortical leads, and the definition of lateral and anterior PFC across subjects are important.

3) There is a notable lack of specificity of the observed effects. In the LFP analysis, every brain region pretty much shows the same thing – and one might wonder if LFP recording of regions outside of this circuit (anterior PFC, lateral PFC, and STN) would also show the same thing – i.e. how specific is the beta desynch and rebound to these cortical-BG regions? The greater anatomical specificity asked for in comment (2) may help address this concern.

4) Figure 5 shows a reduction of firing for both distractor and target, and an earlier uptick for distractor (not sig. at around the 200 or 250 ms mark after the encoding letter is shown). The LFP findings in Figure 3 show that difference starting at about 400 ms (albeit time-resolution limited for this band). And, the coherence lateral PFC – STN for LFP in Figure 5 is after about the 500 ms mark. Some of these timings might be early enough to be "stop", others seem late in the day. There is not a discussion of these timings in relation to putative psychological processes of cortical-BG suppression/gating and in relation to how long it takes to encode/read a word, process the distractor shape, and so forth. Does the timing of the beta 'increase' reflect an abortion/stop process to prevent the distractor word from entering awareness, akin to motor stopping? In the ECoG studies of Swann/Tandon/Aron/Wessel and colleagues, there is an uptick of beta before the time of behavioral stopping. Or is this instead akin to the 'rebound' that occurs in beta activity following movement, that is later? And what are the relative implications of these different interpretations for the wider conclusion, and in relation to the motor stopping literature?

5) It is implied that ⅔ of the clusters responded in a condition-specific declining manner like beta power. This inference would be much stronger if the authors had computed spike-field coupling between spikes and beta phase and used that to dissociate which spikes should be expected to behave as beta did. It would also be helpful for the authors to communicate if there were any location differences in the STN for these two types of clusters.

6) A strong feature of the study is simultaneous LFP/spiking recording, yet there is little discussion (or analysis) of the relation between the two. Further, the findings from the spike data did not provide much insight into their role in non-motor behavior.

---

## [Author Response]

Essential revisions:1) In the Results and the Statistical analysis section, it is unclear whether data are being pooled across subjects (for a 'fixed effects' analysis) or analyzed separately for each subject then carried up to the group level to do statistics across subjects ('random effects' analysis). This has implications for the conclusions – if the former, then it is not possible to generalize beyond the population studied in the paper and this should be explicitly noted in the Results and Discussion section. Further, the reviewers agreed that in the absence of proper random effects analysis, analysis within each subject should be reported, along with the prevalence of the effect across individuals.

We apologize for any lack of clarity in our original description of the methods used to assess any changes in neural activity across participants. To directly answer the reviewers’ question, we do not pool data across participants (fixed effects analysis), and instead we use a random effects analysis at the level of the experimental sessions in order to draw any conclusions that we make. Our main analyses involve computing the changes in spectral power during the task compared to baseline and in computing the difference in mean power between target and distractor conditions. In both comparisons, we test if the distribution of differences across the experimental sessions was significantly different from zero using a permutation procedure where each individual session is a unit of observation. In this case, each session is assigned a value computed as the mean power spectrogram averaged across all trials within that session. Because these session-averages are compared across sessions using a permutation procedure, we believe this represent a random-effects analysis. Furthermore, to demonstrate that this effect is not specific to only one or two participants, we have also included an additional analysis in which we specifically identify how many individual sessions demonstrated at least one electrode contact showing a significant difference between conditions at the individual session level. For this analysis, each individual trial is the unit of observation.

Materials and methods

“To assess overall changes in spectral power regardless of trial type during the task compared to baseline, we performed a random-effects statistical analysis in each brain region across experimental sessions. […] Each of the 200 surrogates was generated by randomly either subtracting the baseline from spectral power during the trial or subtracting the spectral power during the trial from the baseline in each trial, and then recomputing the mean difference in the spectrogram across trials.”

2) The cortical electrode locations were unclear and difficult to ascertain from the general labels, like "anterior PFC". The example participant reconstruction provided in the supplement does little to clarify. While Figure 1—figure supplement 1 has realistic/detailed reconstructions of electrode position, this is just for one subject. More detailed anatomical information about the position of the cortical leads, and the definition of lateral and anterior PFC across subjects are important.

This is a good point, and we agree that having more specific anatomic locations for the cortical electrodes would be a valuable addition to our manuscript. Unfortunately, we did not have access to intraoperative CT to precisely localize electrode contacts during the task. As such, in our initial manuscript, we elected to avoid making specific claims about anatomic locations and decided upon a more conservative approach in which we could only draw conclusions regarding broad brain regions. We have now clarified this point in the revised Materials and methods. However, it is possible to estimate the electrode locations by identifying the specific location of the burr hole in each participant using the post-operative CT co-registered with the pre-operative MRI, and by the fact that all electrode strips were inserted through the burr hole using the same technique and oriented in the same direction. We initially provided this estimation for one participant, as the reviewer notes. However, we have now included the estimation for two participants as well as the reconstructed electrode position image for every participant, projected on to a standard brain. We have also included an additional standard brain plot showing the location of the lateral PFC electrodes that showed a significant difference in beta power between target and distractor trials. Finally, we have also added this point to the revised Discussion, specifically mentioning this limitation to the study.

Materials and methods

“Because we did not have access to intraoperative computed tomography (CT) imaging, we were unable to use patient specific landmarks to accurately localize individual iEEG contacts. […] This procedure therefore resulted in a single spectrogram that we broadly assigned to each of the two cortical brain regions (anterior and lateral PFC) during each trial.”

Results

“Although we did not have access to intra-operative computed tomography (CT), we estimated all anterior and lateral PFC electrode locations on each participant's brain (see Supplementary Information) to determine whether the observed effects were localized to a particular brain region. We found that despite the fact that the anterior PFC, the lateral PFC, and the STN all showed a similar drop in beta power during the task, the beta power within individual participants was focal in nature, although the location of these focal drops was not consistent across participants (Figure 2—figure supplement 1).”

Discussion

“Though the lack of intraoperative CT imaging limited our ability to determine precisely what areas of the lateral PFC were interacting with the STN, our data provide direct evidence linking these structures and their respective roles as participants make these non-motor decisions related to memory.”

3) There is a notable lack of specificity of the observed effects. In the LFP analysis, every brain region pretty much shows the same thing – and one might wonder if LFP recording of regions outside of this circuit (anterior PFC, lateral PFC, and STN) would also show the same thing – i.e. how specific is the betabeta desynch and rebound to these cortical-BG regions? The greater anatomical specificity asked for in comment (2) may help address this concern.

This is also a good point that deserves more attention than that originally provided in the first submission of this manuscript. Because we only placed electrodes in areas we hypothesized to be involved in this behavior, we are not able to unequivocally conclude that other brain regions would not also demonstrate a decrease in beta power during the task. Nevertheless, we believe that the decreases in beta power we observed during the task are focal processes that do not reflect brain-wide changes in activity. Support for this claim stems from the fact that within any given electrode strip that showed a decrease in beta power during the task, not all contacts demonstrated that change when they were analyzed within individual sessions. We have conducted analyses demonstrating this point, and we have added the following paragraph to the results to highlight these focal changes in beta power. We have also added a similar paragraph for the target vs. distractor trial type comparison.

Results

“Although we observed these decreases in beta oscillatory power in both the anterior and lateral PFC by averaging the spectral power across all electrode contacts, we were interested in whether the observed changes represented a focal process or a diffuse cortical phenomenon. […] Conversely, in the anterior PFC, only 6 out of 29 sessions demonstrated at least one contact exhibiting a significant difference in average beta power between target and distractor trials (𝑝 <.05, permutation test).”

4) Figure 5 shows a reduction of firing for both distractor and target, and an earlier uptick for distractor (not sig. at around the 200 or 250 ms mark after the encoding letter is shown). The LFP findings in Figure 3 show that difference starting at about 400 ms (albeit time-resolution limited for this band). And, the coherence lateral PFC – STN for LFP in Figure 5 is after about the 500 ms mark. Some of these timings might be early enough to be "stop", others seem late in the day. There is not a discussion of these timings in relation to putative psychological processes of cortical-BG suppression/gating and in relation to how long it takes to encode/read a word, process the distractor shape, and so forth. Does the timing of the beta 'increase' reflect an abortion/stop process to prevent the distractor word from entering awareness, akin to motor stopping? In the ECoG studies of Swann/Tandon/Aron/Wessel and colleagues, there is an uptick of beta before the time of behavioral stopping. Or is this instead akin to the 'rebound' that occurs in beta activity following movement, that is later? And what are the relative implications of these different interpretations for the wider conclusion, and in relation to the motor stopping literature?

We thank the reviewers for raising this issue. One of the limitations of a task that does not involve explicit motor movements is that we lack behavioral measurements to estimate when and for how long the subjects were engaged with each trial. Nevertheless, as the reviewers note, prior work involving STN recordings during Go-NoGo or stop signal reaction time tasks provides significant insight on the relative timing of neuronal activity related to the execution and inhibition of actions. Notably, Kuhn et al. 2004, Ray et al. 2012, and Alegre et al. 2013 show very similar patterns of beta band timing as those we report here: beta power differences between go and stop trials became significant around 400 ms after the onset of the trial; the response occurred on average 200 ms after that. In addition, the Swann et al. ECoG study mentioned by the reviewers also demonstrated task related changes in beta power around the same time point relative to the go stimulus (100-250 ms after the stop cue, which was on average 233 ms after the go stimulus). The authors of these studies interpret these relative timings to suggest that the increases in beta power occur sufficiently early to be involved in actually stopping the response. Indeed, the calculated stop signal reaction times suggest that the beta power increase are sufficiently early to alter behavior for these studies. Given the fact that the relative timings of beta changes we report in this current study are similar to those seen during the motor response inhibition studies, we believe that the activity is indeed involved in stopping the encoding of the numbers to working memory. Interestingly, the reviewers also raise the possibility that the difference we observe may instead be related to differences in the beta rebound that are often seen following movement. We feel that this is less likely. As explicitly tested in Alegre et al. 2013, these rebound effects tend to occur much later after the execution of the action (1500 ms after the Go stimulus). Moreover, the rebound effects do not occur on the successfully inhibited trials in which no action was executed, and only occur during trials that explicitly involve the execution of an action (correct Go trials and the incorrect Stop trials).

In addition, the motor response studies mentioned above also demonstrate that the beta power differences between motor execution trials and motor inhibition trials continued even after the response was initiated. Furthermore, even on failed trials in which subjects were unable to properly inhibit the erroneous motor response, there were still significant differences between the Go trials and the incorrect STOP trials. This would suggest that an inhibitory signal need not necessarily occur before the initiation of an action (motor or otherwise) to be involved in the inhibition of that action. Indeed, previous work has shown that erroneous movements are often executed with less force than correctly executed movements (Ruiz et al. 2011) and that partial errors in which subjects initiate but then quickly terminate an erroneous response demonstrate different electrophysiological activity than fully executed errors (Cohen and Van Gaal 2014). Even when the brain fails to prevent an action, it may still try to further inhibit its execution. Thus, it is possible that the beta power differences we observe late in the current memory inhibition task are involved in suppressing the encoding or maintenance of a memory even after the encoding process has begun.

We agree with the reviewers that these are important points worth raising, and we have now revised our Discussion to include these points accordingly:

Discussion

“The absence of movement here therefore allows us to specifically examine changes in STN activity during non-motor decisions and show that these changes are similar to those seen during movement inhibition. […] The concept of global suppression has been well described in the context of movement, but more recently has been extended to include even non-motor processes such as working memory (Wesssel et al., 2016a).”

5) It is implied that ⅔ of the clusters responded in a condition-specific declining manner like beta power. This inference would be much stronger if the authors had computed spike-field coupling between spikes and beta phase and used that to dissociate which spikes should be expected to behave as beta did. It would also be helpful for the authors to communicate if there were any location differences in the STN for these two types of clusters.

We thank the reviewers for this suggestion, which we feel has significantly strengthened our manuscript. We have now performed an analysis of spike phase locking in order to determine whether the neuronal clusters that demonstrated decreases in spiking activity similar to the decreases observed in the beta oscillations were locked to those oscillations. We found that indeed they were, and have now included this additional analysis in our Results. Interestingly, the neuronal clusters that exhibited increases in spiking activity during the task did not show evidence for beta phase locking, suggesting that these two types of clusters exhibit different relationships with the ongoing oscillatory activity. We have also included these data in our revised manuscript.

Per the reviewers’ request, we have also added to the manuscript the location of the up and down cells relative to the borders of the STN. There were no significant differences in location between the two clusters.

Results

“Relative to the dorsal and ventral borders of the STN, there were no significant differences in depth between the upward (mean depth 70.4 ± 7.8% ; see Materials and methods) and the downward firing clusters (mean depth 56.2 ± 5.4% ; p = 0.14, unpaired t-test). […] Notably, the neuronal clusters that individually demonstrated increases in spiking activity during the task did not exhibit a significant interaction between LFP phase and spiking events (p >.05, permutation test; Figure 5).

6) A strong feature of the study is simultaneous LFP/spiking recording, yet there is little discussion (or analysis) of the relation between the two. Further, the findings from the spike data did not provide much insight into their role in non-motor behavior.

We agree with the reviewers that the ability to examine the relation between spiking activity and the LFP oscillations would be one of the advantages of simultaneously recording both signals. In our original manuscript, we had not included these analyses because we were specifically interested in whether there was a significant difference in spike phase locking between target and distractor trials. We did not find that this was the case. However, we agree that even this negative result would be important to show, and have now included this in our revised manuscript. In addition, although we did not find that spike phase locking was different between target and distractor trials, we do find evidence that neuronal spiking events are locked to the ongoing oscillation. Indeed, based on the reviewers’ excellent suggestion above, we found that the differences in spike phase locking can distinguish the different neuronal clusters. As noted above, we have included these new analyses in our Results, and have discussed these findings in the revised Discussion.

Regarding the changes in spiking activity in general, the primary insight provided by the spiking data is that individual cells within the STN seem to alter their firing rates during a task the does not involve the gating of movement. We believe this is a valuable contribution to the field, since the basal ganglia in general and the STN in particular have traditionally been implicated in motor movements. Furthermore, some of these cells discriminate between whether or not an item should be encoded into working memory, suggesting that some neurons in the STN participate even in non-motor decisions. Despite these findings, however, we agree that these spiking data alone may not be sufficient for understanding how the STN participates in this process. However, we do feel that they provide an important complement to the larger insights provided by the macroelectrode LFP and intracranial EEG recordings described in this study.

Results

“Moreover, although neuronal clusters exhibiting decreased spiking activity during the task were locked to the LFP beta oscillation, we found no significant differences in beta band spike phase locking between the target and distractor trials (p >.05, permutation test).”

Discussion

**“**Indeed, the observed overall decreases in spiking activity was mediated by only a subset of neurons. […] Hence, while our oscillatory data would implicate the basal ganglia at large in such non-motor decisions that distinguish target and distractor items, the spiking data would suggest that only a subset of STN neurons are involved in this process, and that these neurons do not alter the extent to which they are locked to the ongoing beta oscillations accordingly.”